# Streamlined copper defenses make *Bordetella pertussis* reliant on custom-made operon

Alex Rivera-Millot[1], Stéphanie Slupek[1], Jonathan Chatagnon[1], Gauthier Roy[1], Jean-Michel Saliou [2], Gabriel Billon[3], Véronique Alaimo[3], David Hot [2], Sophie Salomé-Desnoulez[1,4], Camille Locht[1], Rudy Antoine[1✉] & Françoise Jacob-Dubuisson [1✉]

Copper is both essential and toxic to living beings, which tightly controls its intracellular concentration. At the host–pathogen interface, copper is used by phagocytic cells to kill invading microorganisms. We investigated copper homeostasis in *Bordetella pertussis*, which lives in the human respiratory mucosa and has no environmental reservoir. *B. pertussis* has considerably streamlined copper homeostasis mechanisms relative to other Gram-negative bacteria. Its single remaining defense line consists of a metallochaperone diverted for copper passivation, CopZ, and two peroxide detoxification enzymes, PrxGrx and GorB, which together fight stresses encountered in phagocytic cells. Those proteins are encoded by an original, composite operon assembled in an environmental ancestor, which is under sensitive control by copper. This system appears to contribute to persistent infection in the nasal cavity of *B. pertussis*-infected mice. Combining responses to co-occurring stresses in a tailored operon reveals a strategy adopted by a host-restricted pathogen to optimize survival at minimal energy expenditure.

[1] Univ. Lille, CNRS, Inserm, CHU Lille, Institut Pasteur de Lille, U1019- UMR 9017-CIIL-Center for Infection and Immunity of Lille, Lille, France. [2] Univ. Lille, CNRS, Inserm, CHU Lille, Institut Pasteur de Lille, US 41 - UMS 2014 - PLBS, F-59000 Lille, France. [3] Univ. Lille, CNRS, UMR 8516 – LASIRE – Laboratoire de Spectroscopie pour les Interactions, la Réactivité et l'Environnement, F-59000 Lille, France. [4] Bio Imaging Center Lille platform (BICeL), Univ. Lille, Lille, France.
✉email: rudy.antoine@inserm.fr; francoise.jacob@ibl.cnrs.fr

Copper is an essential trace element bioavailable in its soluble $Cu^{2+}$ form in oxygen-rich environments[1]. As the redox potential of the $Cu^{1+}/Cu^{2+}$ pair is exquisitely tuned for enzymatic reactions, most living organisms use copper for oxidation reactions and in electron transfer chains. However, copper excess is highly toxic. In the reducing environment of the cytoplasm, $Cu^{1+}$ displaces iron and other transition metals from their sites in metalloenzymes, causing loss of function and deregulating metal import systems[2,3]. $Cu^{1+}$ also indirectly generates intracellular oxidative stress[4,5]. Because of its toxicity, living organisms have evolved a variety of systems to fend off excess of this metal. As such, ubiquitous copper-specific $P_{IB}$-type ATPases remove copper from the cytoplasm, in partnership with cytoplasmic copper chaperones in both eukaryotes and prokaryotes[1,6,7].

Eukaryotic organisms take advantage of copper toxicity to kill invading microorganisms[8,9]. Predatory protists, such as amoebae, and phagocytic cells of higher eukaryotes import copper for bactericidal purposes[10,11]. In turn, bacteria have developed various resistance mechanisms against copper excess[12,13]. Copper extrusion systems include CopA-type ATPases, as well as CusABCF-type resistance-nodulation-cell division (RND) transporters in Gram-negative bacteria, and other less well-characterized proteins[1,14]. Oxidation of $Cu^{1+}$ in the periplasm is catalyzed by PcoA-type multi-copper oxidases (MCO)[15]. Specific metallochaperones also contribute to copper homeostasis as part of export systems or by copper sequestration[16,17].

Environmental bacteria are likely to encounter occasional copper excess or predation by protists. Therefore, they have developed large resistance arsenals against copper intoxication. Conversely, bacteria with restricted ecological niches, such as pathogens or commensals, appear to have fewer copper homeostasis genes[18]. Here, we investigated copper homeostasis in the whooping cough agent *Bordetella pertussis*, a predominantly extracellular pathogen whose sole natural niche is the human upper respiratory tract[19]. Unlike its relative *Bordetella bronchiseptica*, whose lifestyles alternate between the environment and a mammalian host, *B. pertussis* is host-restricted and was thus used here as a model to study how genomic reduction by niche specialization affects copper homeostasis at the host–pathogen interface. We discovered that *B. pertussis* has eliminated most resistance systems typically found in Gram-negative bacteria. Its only line of defense consists of an original operon to withstand copper excess and peroxides, both encountered in phagocytic cells.

## Results

**Loss of copper homeostasis systems in *B. pertussis*.** Comparative genomic analyses revealed that the exclusively human pathogen *B. pertussis* and its close relative *B. bronchiseptica* share common sets of copper-homeostasis genes, including genes coding for a CopA ATPase, a PcoA-PcoB-type MCO system, and CopZ, CopI, and CusF copper chaperones, but both lack *cusABC* genes[18]. A copper storage protein (*csp*) gene is present in the *B. bronchiseptica* but not in the *B. pertussis* genome[18]. Both species also have genes coding for CueR-type regulators and CopRS-type two-component systems that control copper homeostasis genes in other bacteria[20,21].

Nevertheless, growth of the two species in the presence of copper revealed distinct phenotypes (Fig. 1). After a prolonged lag, Cu-treated *B. pertussis* started growing at the same rate as the untreated culture, suggesting that copper has a bacteriostatic effect, whereas growth of Cu-treated *B. bronchiseptica* started at the same time as the control but at a slower pace, indicating a rapid adaptation to copper-rich conditions. The apparent resistance of both species to high concentrations of copper is misleading and due to growth medium compounds chelating the metal ion, in particular Tris, amino acids, and glutathione, which markedly reduces its effective concentration. We hypothesized that *B. pertussis* started growing after the toxic $Cu^{1+}$ ion resulting from the reduction of $Cu^{2+}$ by ascorbate, another component of the growth medium, was re-oxidized thanks to strong oxygenation of the cultures. This was confirmed by the observation that the addition of fresh ascorbate after a few hours to *B. pertussis* cultured in the presence of 5 mM copper prevented bacterial growth (Supplementary Figure S1).

RNA sequencing (RNA-seq) analyses of *B. pertussis* grown in the presence of 2 mM $CuSO_4$ showed altered expression of a number of stress response and metabolic genes and of a few Cu-specific genes (Fig. 2a, b and Supplementary Data S1). They include *bp2860* and *bp2722* that code for $P_{IB}$-type ATPases, the latter being Zn-specific[22], and *bp1727*, *bp1728*, and *bp1729* that form an operon (Supplementary Figure S2) and code for a CopZ-like copper chaperone, a putative glutaredoxin and a putative oxidoreductase, respectively. Copper-repressed *bp2921* and *bp2922* were not further investigated here. A 30-min Cu treatment of *B. pertussis* confirmed up-regulation of *bp2860* and *bp1727-bp1728-bp1729* (Fig. 2c and Supplementary Data S2). In contrast, *bp3314-bp3315-bp3316* coding for CopI, PcoA, and PcoB homologs were hardly expressed and not regulated by copper in *B. pertussis*, in contrast to their orthologues in *B. bronchiseptica* (Fig. 2b, d and Supplementary Data S3). Proteomic analyses of the two organisms in the presence of copper showed increased production of the proteins coded by the upregulated genes in both bacteria, except for the membrane protein CopA and the small protein CopZ (Fig. 2b, Supplementary Data S4 and S5). Because the size of CopZ might have hampered its detection in global proteomic analyses, we analyzed extracts of copper-treated *B. pertussis* by denaturing electrophoresis and mass fingerprinting of the putative CopZ protein band (Fig. 3). This confirmed that CopZ is strongly overproduced by *B. pertussis* in the presence of copper.

To investigate the role of copper homeostasis genes, we inactivated *bp3314-bp3315-bp3316* (*pcoA* operon), *bp0157-bp0158* (*copRS*), *bp2860* (*copA*) and *bp1727* (*copZ*) in *B. pertussis*. Except for the latter, growth of those mutants was affected by the addition of copper to the medium to the same extent as that of the parental strain (Supplementary Figure S3). In all available *B. pertussis* genomes, three IS481 sequences are present upstream of *bp3314* and one is located within the 5' end of *bp2860* (Supplementary Figures S4 and S5), which most likely inactivated the *pcoA* and *copA* operons, as evidenced by their low expression levels both in the presence and in the absence of copper (Fig. 2b). In contrast, the *copA* and *pcoA* operons are strongly upregulated by copper and probably functional in *B. bronchiseptica*. Altogether thus, *B. pertussis* has considerably streamlined its response to copper excess, maintaining only one system functional.

**Function of the *copZ* operon.** A strong effect of copper excess on growth was observed with the Δ*bp1727* (*copZ*) mutant, and therefore we focused on that operon. We generated two additional knock-out mutants (Δ*bp1727-bp1728-bp1729* and Δ*bp1728-bp1729*) and measured their growth in the presence or absence of copper. Copper excess caused a major growth defect to two of the mutants, Δ*bp1727*, and Δ*bp1727-bp1728-bp1729*, and the phenotype of the latter was complemented by ectopic expression of the operon (Fig. 4a). Glutathione is a major player in Cu homeostasis[23]. *B. pertussis* cannot synthesize glutathione but actively imports it, as indicated by the high expression levels

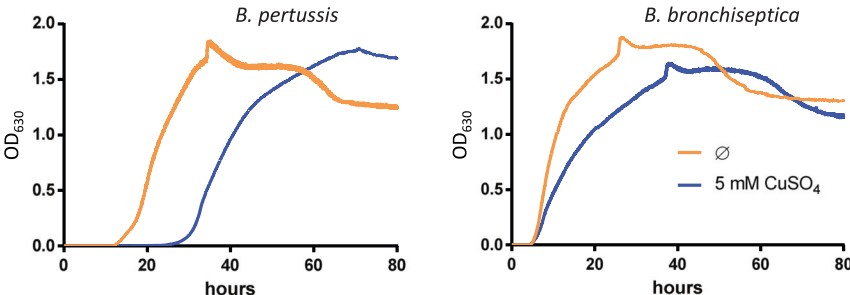

**Fig. 1 Effect of copper on *B. pertussis* and *B. bronchiseptica* growth.** Growth curves of *B. pertussis* BPSM and *B. bronchiseptica* RB50 in SS medium supplemented (blue curves) or not (orange curves) with 5 mM $CuSO_4$ are shown. The turbidity of the cultures was measured using an Elocheck device. The values shown in ordinate (absorbency units) do not correspond to optical density measurements classically obtained with a spectrometer, since the Elocheck instrument uses distinct pathlength and wavelength.

a

b

| | | *B. pertussis* | | | | | *B. bronchiseptica* | | | |
|---|---|---|---|---|---|---|---|---|---|---|
| | | Transcriptomics (RPKM) | | Proteomics (Spectral count) | | | Transcriptomics (RPKM) | | Proteomics (Spectral count) | |
| Protein | Gene | standard | Copper | standard | Copper | Gene | standard | Copper | standard | Copper |
| CopZ* | bp1727 | 162 | 5550 | 0 | 2 | bb3012 | 338 | 2145 | 0 | 0 |
| PrxGrx | bp1728 | 288 | 5059 | 1 | 6 | bb3011 | 478 | 664 | 10 | 22 |
| GorB | bp1729 | 18 | 241 | 0 | 57 | bb3010 | 42 | 23 | 1 | 8 |
| CopA | bp2860 | 2 | 74 | 0 | 0 | bb1180 | 89 | 373 | 0 | 0 |
| CopI* | bp3314 | 1 | 1 | 0 | 0 | bb4581 | 7 | 145 | 0 | 17 |
| PcoA | bp3315 | 2 | 2 | 0 | 0 | bb4580 | 8 | 183 | 0 | 52 |
| PcoB | bp3316 | 1 | 2 | 0 | 0 | bb4579 | 8 | 166 | 0 | 18 |
| Blue copper protein * | bp0156 | 34 | 26 | 0 | 4 | bb4473 | 44 | 222 | 0 | 17 |
| CopS | bp0157 | 4 | 5 | 0 | 0 | bb4472 | 12 | 12 | 1 | 0 |
| CopR | bp0158 | 13 | 12 | 0 | 0 | bb4471 | 25 | 27 | 1 | 1 |
| CueR | bp1726 | 78 | 181 | 0 | 2 | bb3013 | 138 | 208 | 0 | 0 |
| CusF* | bp3088 | 820 | 989 | 0 | 0 | bb0177 | 160 | 144 | 0 | 0 |
| Csp* | | | | | | bb2842 | 7 | 8 | 0 | 0 |
| CusABC | | | | | | | | | | |

c

d

**Fig. 2 Copper regulation of homeostasis systems in *B. pertussis* and *B. bronchiseptica*.** (**a**, **c**, and **d**) RNAseq analyses of *B. pertussis* grown for 16 h in the presence of 2 mM $CuSO_4$ in SS medium (**a**), *B. pertussis* grown in SS medium and treated for 30 min with 2 mM of $CuSO_4$ (**c**), and *B. bronchiseptica* grown for 12 h in the presence of 2 mM $CuSO_4$ (**d**). Comparisons were made with bacteria grown in standard conditions. Each gene is represented by a dot. The x and y axes show absolute levels of gene expression in reads per kilobase per million base pairs (RPKM) in standard and copper conditions, respectively. The genes indicated in blue indicate genes of interest with the strongest regulation factors. The full sets of data are shown in Supplementary Tables S1, S2 and S3. (**b**) Summary of the transcriptomic and proteomic analyses performed after growing bacteria as in (a) and (c) in medium supplemented with 2 mM $CuSO_4$. Standard culture conditions were used for comparisons. * indicates small proteins difficult to detect by global proteomic approaches.

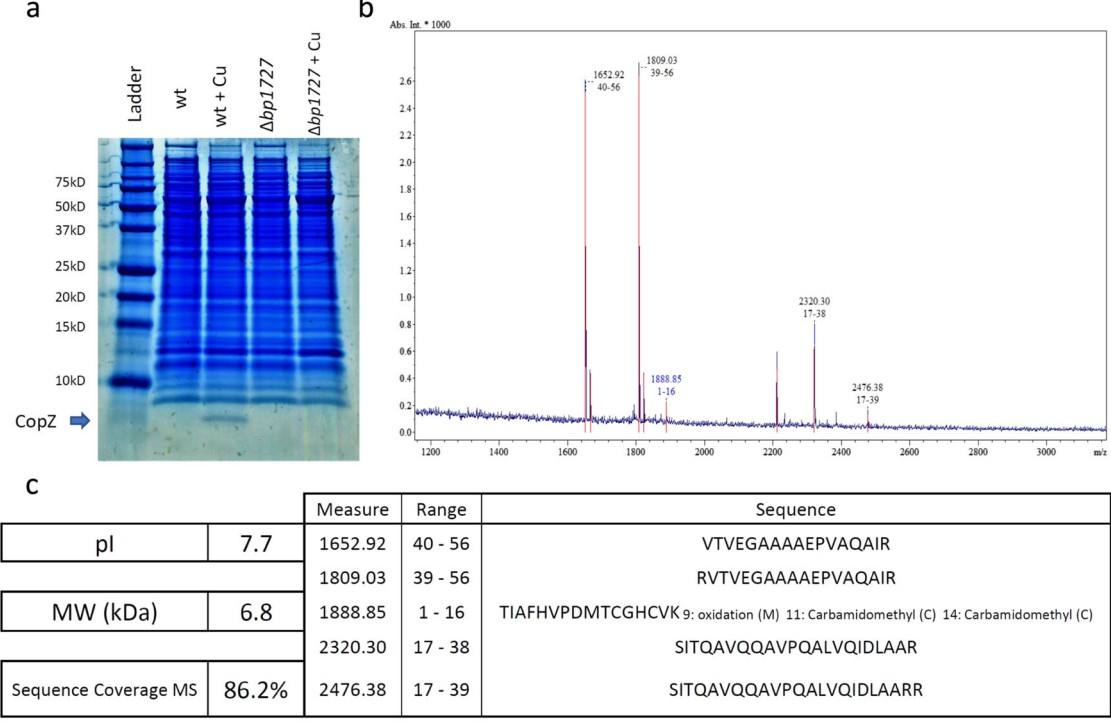

**Fig. 3 Identification of CopZ in *B. pertussis*. a** Lysates of wild type *B. pertussis* (wt) or the deletion mutant (Δ*bp1727*) grown in SS medium supplemented (+ Cu) or not with 2 mM CuSO₄ were subjected to SDS-PAGE electrophoresis in Tris-tricine gels for the detection of CopZ. The gel was stained with colloidal Coomassie blue dye. CopZ migrates below the 10 kDa band of the markers. **b, c** The protein band was cut from the gel, and CopZ was identified by mass fingerprinting analyses. The m/z ratios of the peptides identified by MALDI-TOF are shown in panel b, and the sequences of the peptides are in **c**. This experiment complements the proteomic analyses, because the small size of CopZ hampered its detection in global proteomic approaches (see Fig. 2b and Supplementary Data S4).

of the glutathione-specific ABC transporter genes *bp3828-3831* in all conditions (Supplementary Data S1 and S2). The growth defect of *B. pertussis* in the presence of copper was exacerbated in the absence of glutathione in the culture medium, supporting its role for copper tolerance in *B. pertussis* (Fig. 4b). Altogether the growth data show that CopZ, encoded by the first gene of the operon, plays a major role in copper resistance, unlike the other two proteins.

In other bacteria, CopZ binds $Cu^{1+}$ and transfers it to an ATPase partner for export[24]. Inductively Coupled Plasma Atomic Emission Spectroscopy (ICP-AES) analyses were performed with purified CopZ incubated with Cu or with Fe as a control. Three Cu ions and one Fe ion were bound per CopZ monomer, respectively (Fig. 5a). We reasoned that in both cases one ion was chelated by the 6-His tag used for recombinant CopZ purification. Therefore, subtraction of a Cu ion yields a ratio of 2 Cu per CopZ monomer. This value is consistent with crystal structures of homologs (pdb accession numbers 6FF2 and 2QIF) showing CopZ dimerization mediated by 4 Cu ions bound by conserved Cys and His residues[25] (Supplementary Figure S6). Thus, CopZ is a copper-binding protein, which in the absence of a functional CopA in *B. pertussis* most likely sequesters cytosolic copper to counter its toxicity.

Homology searches indicated that *bp1728* and *bp1729* code for a chimeric peroxiredoxin-glutaredoxin protein[26] and a glutathione reductase, respectively. Orthologues of these enzymes were shown to reduce peroxides at the expense of glutathione and to regenerate glutathione using NAD(P)H, respectively[27,28]. Using purified recombinant BP1728 and BP1729 proteins in in vitro enzymatic assays[27,28], we confirmed that these proteins are a glutathione-dependent peroxidase and a glutathione

reductase, respectively (Fig. 5b–d). We thereafter called them PrxGrx (http://peroxibase.toulouse.inra.fr/) and GorB.

To test whether the PrxGrx-GorB system protects bacteria against peroxides, we determined mutant survival after oxidative stress[29]. A 30-min exposure killed each of the three mutant strains to a greater extent than their parental strain, and the phenotype of the three-gene mutant was complemented by ectopic expression of the operon (Fig. 4c). The phenotype of the Δ*copZ* mutant indicated that, in addition to PrxGrx-GorB, CopZ also protects *B. pertussis* against reactive oxygen species (ROS). Together, the three proteins might endow *B. pertussis* with some capability to fend off both copper and peroxides (Fig. 5e).

As a human-restricted pathogen, *B. pertussis* may simultaneously encounter oxidative and copper stresses in phagosomes of phagocytic cells. We thus tested the role of the operon in intracellular survival within human macrophages and in a murine model of lung and nasal colonization. The deletion mutant was killed faster by macrophages than its parental strain (Fig. 4d). We also subjected the *copZ* and *prxgrx-gorB* mutants to this assay. Both were killed faster than the wild type (wt) strain in macrophages, especially at the 1-hour time point (Supplementary Figure S7). This shows that both CopZ and the Prxgrx-GorB system contribute to survival early after phagocytosis. Finally, we tested the effect of deleting the entire operon in an animal model of colonization. The profiles of colonization in the lungs over time by the mutant and the wt strains were not significantly different, although at the late time point (21 days) more animals appeared to have cleared the mutant bacteria. Furthermore, the deletion mutant was cleared more quickly from nasopharynxes than the parental strain (Fig. 4e, f).

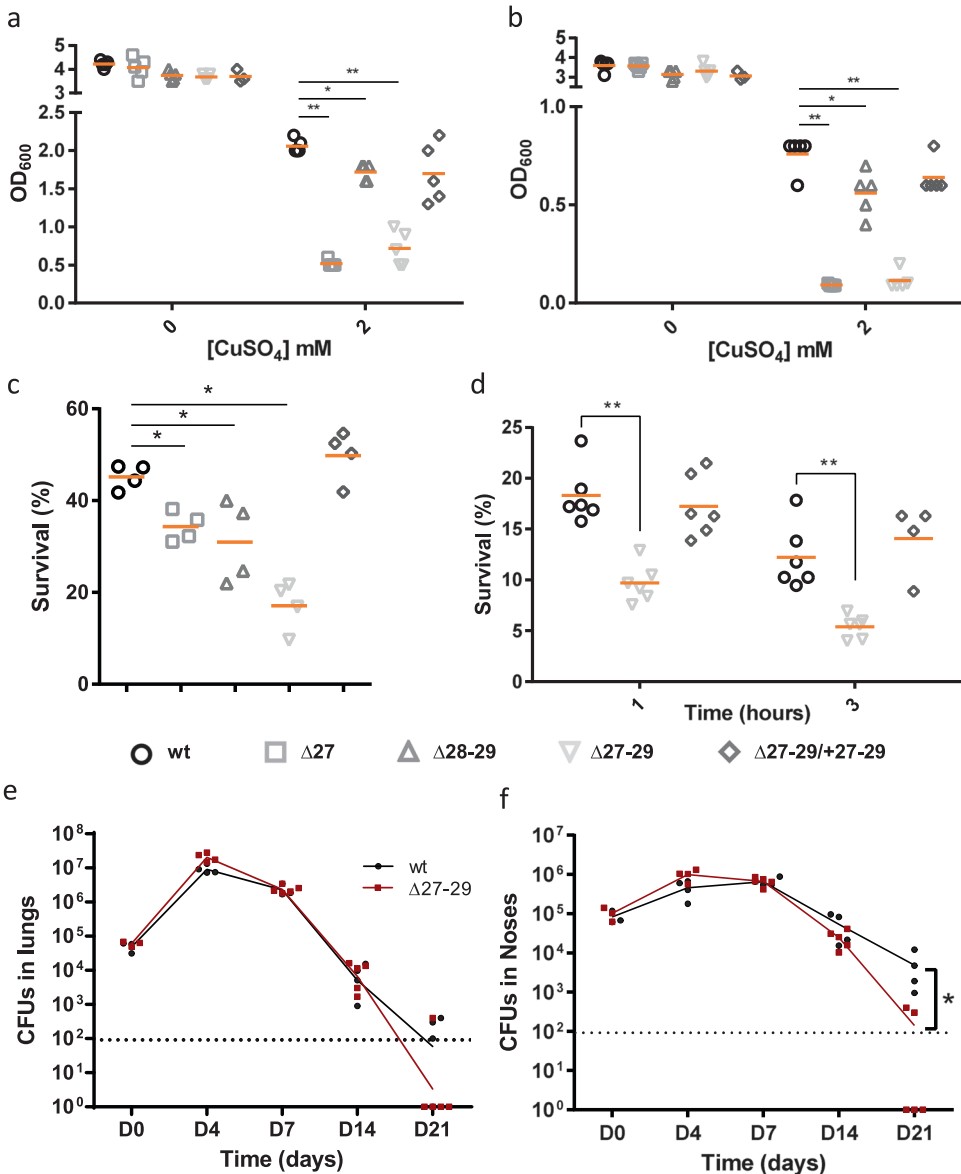

**Fig. 4 Role of the operon in *B. pertussis* and in host–pathogen interactions. a**, **b** Growth yields of *B. pertussis* after 24 h in SS medium (**a**) or in SS medium devoid of glutathione (**b**), supplemented or not with 2 mM $CuSO_4$. The cultures were inoculated at initial $OD_{600}$ values of 0.1, and after 24 h the $OD_{600}$ was determined. Note the different y axis scales between (**a**) and (**b**). wt, wild type (parental) strain; Δ27, Δ28-29 and Δ27-29, KO mutants for *bp1727* (*copZ*), *bp1728-bp1729* (*prxgrx-gorB*) and the three genes, respectively. The various strains are represented using different symbols and shades of gray. Δ27-29/+27-29 represents the latter mutant complemented by expression of the operon at another chromosomal locus. The horizontal orange lines represent mean values. **c** Survival of the same strains to an oxidative shock of 30 min. **d** Intracellular survival of *B. pertussis* in THP1 macrophages. The horizontal orange lines represent mean values. See Supplementary Figure S6 for the data of the individual *copZ* and *prxgrx-gorB* KO mutants. **e**, **f** Colonization of mice lungs (**e**) and nasopharynxes (**f**) after nasal infections with the parental strain and the *copZ-prxgrx-gorB* KO mutant. The numbers of bacteria are indicated for each mouse and organ (black circles: parental bacteria, red squares: mutant). The lines connect the geometric means of the counts at each time point, and the dotted lines indicate the thresholds of detection. For all the assays, statistical analyses were performed using two-tailed Mann-Whitney tests (*, $p < 0.05$; **, $p < 0.005$). For panels **a** and **b**, 5 biologically independent samples were used. For panels **c** and **d**, 4 and 6 biological samples were used, respectively. For panels **e** and **f**, 5 and 4 animals per time point were used for the KO and wt strains, respectively.

**Copper regulation of the *copZ-prxgrx-gorB* operon.** Quantitative RT-PCR (qRT-PCR) on *prxgrx* showed a 30-fold up-regulation of the operon by copper, even at micromolar concentrations (Fig. 6a and Supplementary Figure S8). qRT-PCR experiments on the very small *copZ* gene were unsuccessful, as no qRT-PCR primer pairs were found to work. Inactivation of *bp1726*, in antisense orientation of the *copZ-prxgrx-gorB* operon and coding for a putative CueR regulator[20], resulted in a markedly decreased but not abolished transcriptional response of *prxgrx* to copper (Fig. 6a). The three genes are co-transcribed

(Supplementary Figure S2) and up-regulated by copper (Fig. 2 and Supplementary Data S1 and S2), and we showed that the second is controlled by CueR. We can thus reasonably assume that the entire operon is under the control of CueR. Interestingly, though, the remaining copper-induced up-regulation of *prxgrx* in the *cueR* KO mutant indicates that the operon may also be controlled by a second system directly or indirectly activated by copper.

We determined the transcriptional start site of the operon by 5′ rapid amplification of cDNA ends (5′ RACE). The 5′ UTR is 61-

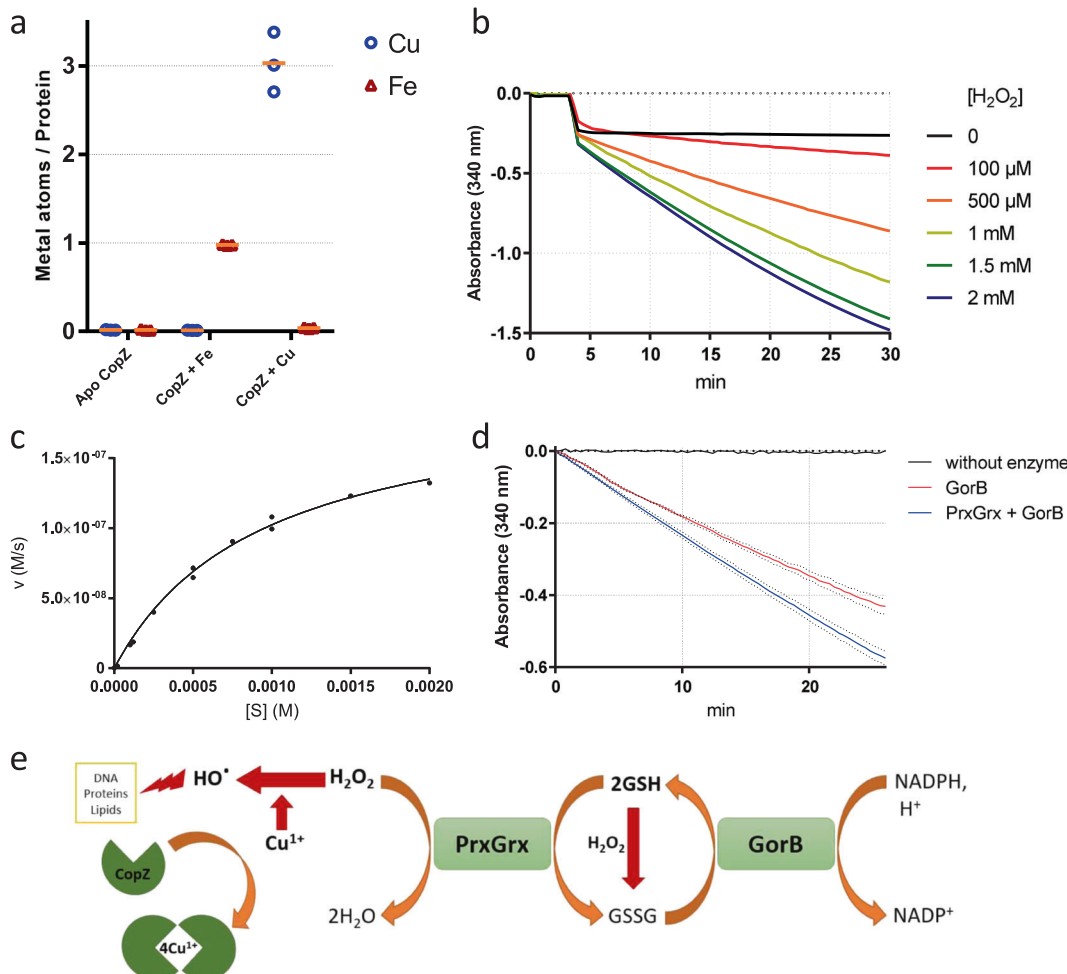

**Fig. 5 Activities of the proteins coded by the *bp1727-1728-1729* (*copZ-prxgrx-gorB*) operon. a** The metal/protein ratios of recombinant CopZ were determined by ICP-AES. The apo form of CopZ was incubated or not with iron (gray) or copper (black), and both ions were measured. Three independent samples were processed. **b** Oxidation of NADPH by recombinant GorB over time. $H_2O_2$ was added at the indicated concentrations to generate the glutathione disulfide (GSSG) substrate of GorB, before adding the enzyme. The reaction was followed by the decrease of absorbency at 340 nm. **c** Plot of the reaction rate as a function of substrate concentration. A $k_{cat}/K_m$ value of $323{,}000 \pm 46{,}000 \, M^{-1}s^{-1}$ for GSSG was estimated based on Michaelis–Menten kinetics. One or two measurements were performed at each substrate concentration. **d** Effect of recombinant PrxGrx on the rate of oxidation of NADPH by GorB. The red and blue curves show reaction rates when GorB was present alone or when both enzymes were present, respectively. $H_2O_2$ was added last as a substrate of the first reaction. However, as $H_2O_2$ also generates GSSG, which initiates the second reaction, PrxGrx activity was detected by the increased rate of the reaction when both enzymes are present, and no enzymatic constants could be determined. (**e**) Schematic representation of the functions of the three proteins. By chelating $Cu^{1+}$, CopZ prevents the ion from generating hydroxyl radicals through the Fenton reaction. $H_2O_2$ is reduced to $H_2O$ through the activity of PrxGrx, at the expense of reduced glutathione (GSH). The product of that reaction, glutathione disulfide, is reduced through the activity of GorB at the expense of NADPH.

nucleotides long, and putative CueR binding sites were identified within the promoter, as described for *Escherichia coli*[30] (Supplementary Figure S5). Attempts to produce a recombinant CueR protein in *E. coli* to be used for electrophoretic migration shift assays (EMSA) were unsuccessful, suggesting that its over-production may be toxic to *E. coli*.

We then turned to human macrophages to determine if the operon is up-regulated upon phagocytosis. It was shown earlier that the ATP7A Cu-transporting ATPase is overexpressed and localizes to phagosomes in activated murine macrophages[9]. We thus reasoned that copper might trigger expression of the *copZ-prxgrx-gorB* operon in *B. pertussis* early after phagocytosis. By using epifluorescence microscopy, we showed that differentiation and activation of THP1 cells cause overproduction of ATP7A. In addition, the Cu transporter partially redistributes from a mainly ER/Golgi localization in resting cells to a more punctuated localization corresponding to cytosolic vesicles (Supplementary

Figure S9). Endosomes are thus likely to be loaded with copper. Then, to visualize the activation of the operon in intracellular bacteria, we introduced a transcriptional fusion between the promoter of the operon and an mRFP1 reporter gene in *B. pertussis* and put the bacteria in contact with differentiated THP1 cells deprived of copper or supplemented with copper. After one hour of contact, fluorescence of bacteria engulfed by copper-replete macrophages was significantly more intense than in copper-deprived macrophages (Fig. 7a, b). The rapid induction of the *copZ-prxgrx-gorB* promoter indicates that copper is an early signal for intracellular bacteria, and that CopZ-PrxGrx-GorB serves as a defense system in early phagosomes. This is supported by the role of that system for bacterial survival at early time points after phagocytosis (Fig. 4d).

**Regulation by peroxides.** As shown above, residual regulation of *prxgrx* by copper was observed in the *cueR* KO strain (Fig. 6a).

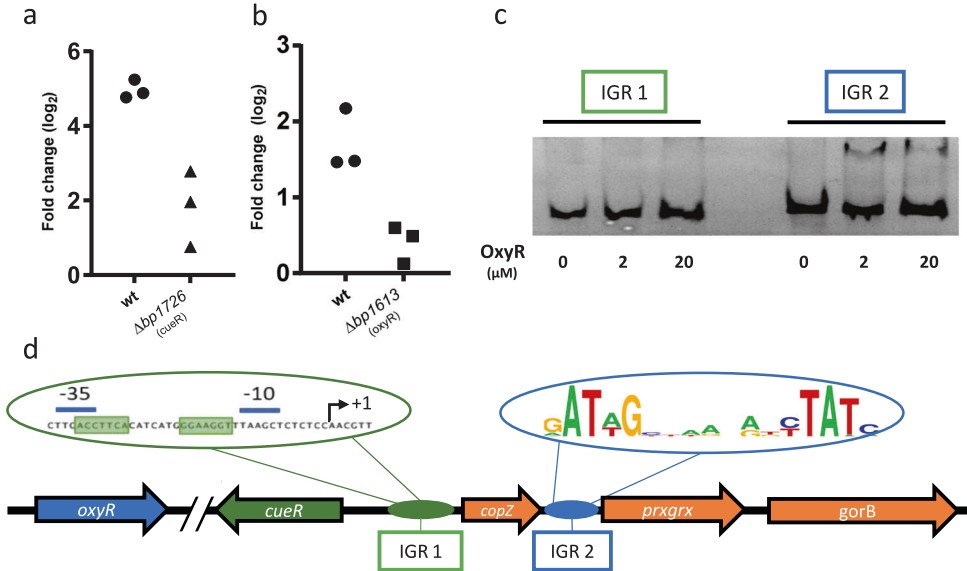

**Fig. 6 Regulation of the operon. a, b** qRT-PCR analyses of the parental and the *cueR* KO strains treated for 30 min with 2 mM CuSO$_4$ (**a**) or 10 mM H$_2$O$_2$ (**b**), showing the expression levels of *prxgrx* relative to untreated controls. Data were normalized with the housekeeping gene *bp3416*. Three biological replicates were performed. (**c**) EMSA with recombinant OxyR and DNA fragments of the *cueR-copZ* and *copZ-prxgrx* intergenic regions, IGR 1 and IGR 2, respectively. The uncropped gel is shown in Supplementary Figure S10. (**d**) Schematic representation of the locus, with sequences of the putative CueR and OxyR boxes. The site of transcription initiation was determined by 5'RACE (Supplementary Figure S5). The putative OxyR binding sites were identified by their similarity with the *E. coli* consensus sequences, and alignments of the *Bordetella* and *Achromobacter* sequences were used to build the consensus motif (Supplementary Figure S11).

Since copper excess generates oxidative stress[4], we hypothesized that a ROS-sensing regulator might be involved in this residual control of *prxgrx*. We examined all known prokaryotic PrxGrx homologs and searched for putative regulatory genes in the corresponding loci. An *oxyR* gene was found directly adjacent and in antisense orientation in 19.4% of cases (Supplementary Data S6). OxyR regulators have been shown to control bacterial responses to peroxides[31,32].

Treatment of *B. pertussis* with H$_2$O$_2$ resulted in a 4-fold upregulation of *prxgrx*, which was abolished by inactivation of the putative *oxyR* gene, *bp1613* (Fig. 6b). Using EMSA, we detected a protein-DNA complex between recombinant OxyR and the *copZ-prxgrx* intergenic region but not the *cueR-copZ* intergenic region (Fig. 6c and Supplementary Figure S10). Putative OxyR binding boxes can be found in the *copZ-prxgrx* intergenic region (Supplementary Figure S11). Thus, two transcriptional activators control the expression of this operon in response to distinct signals (Fig. 6d). The three genes are strongly up-regulated by copper through CueR, and in addition, the last two genes are more modestly up-regulated by peroxides through OxyR.

Finally, we searched β proteobacterial genomes for similar composite operons to determine the phylogenetic context in which they arose. Out of 86 bacterial genera, 16 harbor a *prxgrx* gene, of which 11 have *prxgrx-gorB* operons. Only *Bordetella* and *Achromobacter* have complete *copZ-prxgrx-gorB* operons. The latter two genera encompass pathogenic or opportunistic species. Occurrence of this operon thus appears to be linked to eukaryotic cell-associated lifestyles.

## Discussion

Copper is both an essential and toxic metal for living organisms, including bacteria, yet copper homeostasis has been studied only in a few model organisms. We describe here that *B. pertussis*, a host-restricted, mostly extracellular bacterium that lives on human mucosal surfaces, has considerably streamlined its

resistance against copper intoxication. *B. pertussis* mainly relies on a composite, three-protein system strongly upregulated by copper to counter both copper excess and peroxide stress. This system evolved from an environmental bacterial ancestor by diverting a ubiquitous cytoplasmic metallochaperone for copper sequestration and recruiting genes involved in protection against oxidative stress in an operon regulated by copper. This assembly enables the bacterium to mount a strong response against two stresses simultaneously encountered in the endosomal compartment. Molecular traces of this evolution can be found in the OxyR-mediated regulation of the last two genes of the operon in response to peroxide stress. Persistence of this ancestral control may be useful to a strictly aerobic bacterium when it resides outside of the phagosomal environment, as respiration also generates ROS[33].

Genomic, transcriptomic and proteomic analyses demonstrated the loss in *B. pertussis* of most well-known resistance mechanisms against copper excess. This contrasts with other respiratory tract pathogens such as *Mycobacterium tuberculosis*, *Pseudomonas aeruginosa* and *B. bronchiseptica*[34,35]. Unlike *B. pertussis, M. tuberculosis* is a genuine intracellular bacterium that resides within macrophages and other cell types, *P. aeruginosa* can occupy various niches including environmental milieus, and *B. bronchiseptica* alternates between pathogenic and environmental life cycles[36]. In humans, copper is not free but complexed with proteins such as ceruloplasmin or metallothionein. Therefore, *B. pertussis* is unlikely to experience copper excess, except when it is engulfed by phagocytic cells, a fate it actively tries to avoid[37,38]. *B. pertussis* could thus afford the loss of most defense mechanisms against copper intoxication, because of its specialization to a restricted niche.

*B. pertussis* is mainly an extracellular pathogen, with limited survival rates upon phagocytosis[39]. Therefore, it may be surprising that it has nonetheless conserved an active-defense system. Nevertheless, *B. pertussis* has been shown to occasionally reside within phagocytic cells[39–41]. We propose thus that

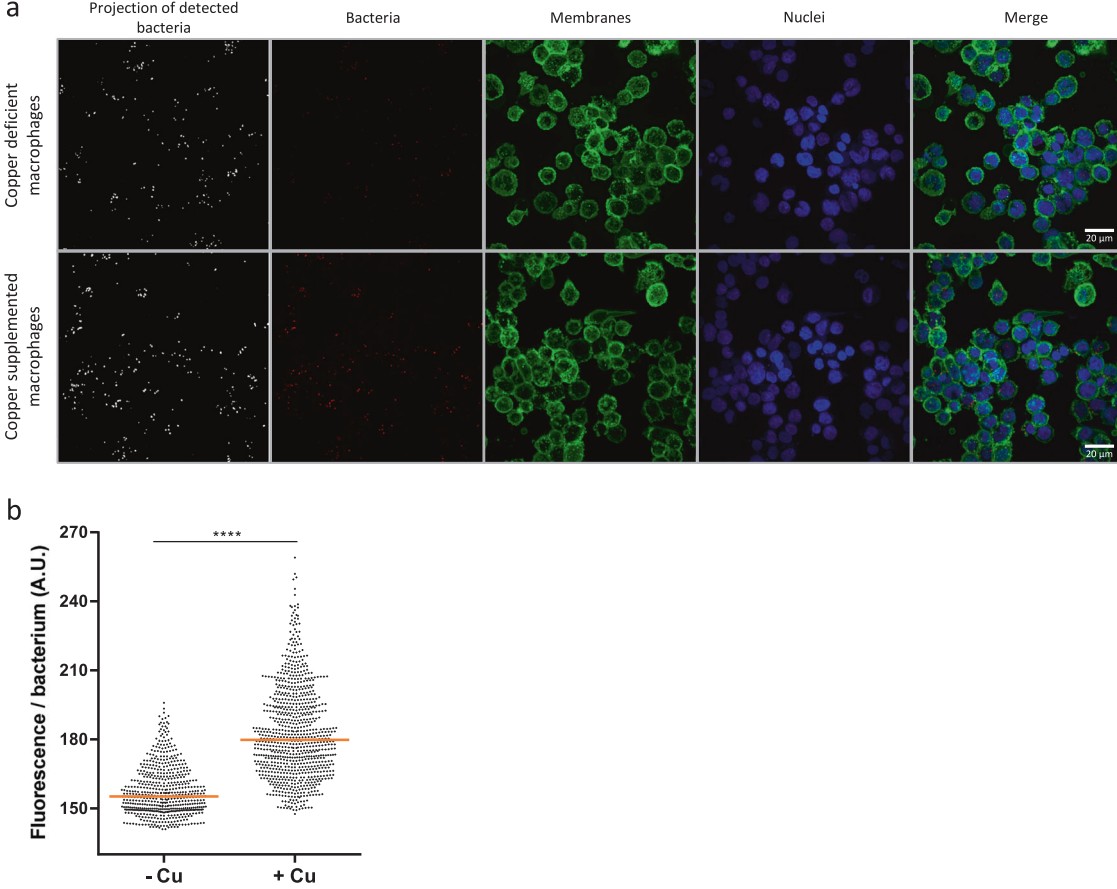

**Fig. 7 Upregulation of the operon in macrophages. a** Representative images of THP1 macrophages having engulfed *B. pertussis* harboring the *mRPF1* gene under the control of the *copZ-prxgrx-gorB* operon promoter. Macrophages were either starved of copper using a chelator or treated with copper chloride prior to contact with bacteria. The bacteria are red, cell membranes are green, and nuclei are blue. In the two fields shown at the left, the bacteria were projected as objects, showing that similar numbers are present in both conditions. All image panels are shown at the same magnification. **b** Levels of fluorescence (arbitrary units) of intracellular bacteria in the two conditions. Statistical analyses were performed using a two-tailed Student T-test (****, $p < 0.0001$). The horizontal orange lines represent median values (n > 500 in both cases).

*B. pertussis* adopts a two-pronged strategy with respect to phagocytic cells while limiting energy expenditure. The major arm of this strategy uses immunomodulatory virulence factors to target neutrophils and macrophages by disabling, delaying or dysregulating their responses to infection[37,38,42–45]. The second arm is to overproduce specific proteins, including those identified here, that enable bacteria to survive to some extent within specific compartments in macrophages, probably with early endosomal characteristics[39], while those phagocytes face the virulence factors produced by extracellular bacteria. Absence of the *copZ-prxgrx-gorB* operon had little effect on *B. pertussis* colonization of the lungs but shortened colonization of the nasal cavity of mice. Interestingly, in murine nasal tissues, a large influx of immune cells occurs after one to two weeks of infection, and a sizeable proportion of bacteria appear to reside within immune cells in that organ (V. Dubois, personal communication). This might contribute to explain the detrimental effect of inactivating the operon on nasal cavity colonization.

Copper and peroxide stresses are interlinked, as copper excess indirectly causes oxidative stress[4]. Placing the *copZ*, *prxgrx*, and *gorB* genes under copper control is thus an efficient way of fighting co-occurring threats, especially considering that CopZ also protects the bacterium from peroxide stress. This activity most likely stems from CopZ's ability to form redox-sensitive Cys-mediated dimers[46]. This property is reminiscent of the eukaryotic CopZ homolog, Atox1, both an anti-oxidant protein and the copper chaperone partner of the CopA homologs ATP7A/B[47,48]. In *B. pertussis*, CopZ sequesters copper but no longer participates in its export. In some bacteria, these two functions are performed by distinct CopZ paralogues[17].

Pathogenic Bordetella species are thought to have derived from environmental ancestors[49]. The *copZ-prxgrx-gorB* operon was only found in two closely related genera, *Bordetella* and *Achromobacter*, arguing that it originated in a common environmental ancestor. Known *Achromobacter* species are environmental bacteria or opportunistic pathogens. The need to survive predation by protists, which employ killing mechanisms similar to those of macrophages, including the use of copper[10], exerts a strong selective pressure on environmental bacteria. This selective pressure most likely accounts for the panoply of resistance systems found in *Achromobacter*[18], which most likely were already present in the environmental ancestor of *Achromobacter* and *Bordetella*. Persistence in ameba could constitute one of the evolutionary steps allowing *Bordetellae* to infect mammals. *B. bronchiseptica* exemplifies an intermediate in this evolution, as this respiratory pathogen of mammals which is able to multiply in amoebae and potentially uses them as transmission vectors has conserved ancestral genes for intracellular persistence[36,50]. As a host-restricted pathogen, *B. pertussis* has traded most of this ability for a lighter coding potential. *B. pertussis* has a number of

cuproproteins[18] and therefore requires copper for growth. It might thus also use CopZ for nutritional passivation[51], i.e., as a copper store for future needs.

## Methods

**Culture conditions.** Modified Stainer-Scholte (SS) liquid medium and Bordet-Gengou blood-agar (BGA) medium were used with the appropriate antibiotics to grow *B. pertussis* BPSM and its derivatives. As copper ions are Lewis acids and *B. pertussis* requires buffered medium close to pH 7, a stock solution (62 mM of $CuSO_4$, 100 mM Tris, pH 5.5) in the fraction A of SS medium was used to supplement the medium with copper. The control cultures were prepared by adding the same volumes of 100 mM Tris-HCl (pH 5.5) to SS medium. To record growth curves, the optical density at 630 nm was continuously measured using an Elocheck device (Biotronix). For growth yields, the *B. pertussis* cultures were started at $OD_{600}$ of 0.1, and after 24 h of growth optical density values at 600 nm ($OD_{600}$) were determined using an Ultrospec 10 spectrophotometer (Biochrom) with five biological replicates for each condition and strain. Glutathione was omitted from the culture medium where indicated. The turbidities measured using the Elocheck and Ultrospec apparatuses, which use different wavelengths and pathlengths, cannot be directly compared. For transcriptomic, proteomic and qRT-PCR experiments, the *B. pertussis* cultures were started at $OD_{600}$ of 0.1 to reach $OD_{600}$ values of 1.5 after 16–20 h, and *B. bronchiseptica* cultures were started at $OD_{600}$ of 0.08 to reach $OD_{600}$ values of 1.5–1.8 after 10–12 h. To subject *B. pertussis* to oxidative stress for qRT-PCR analysis, the bacteria were grown to $OD_{600}$ of 1.5, and 10 mM $H_2O_2$ was added to the cultures for 30 min. To measure survival to oxidative shock, the bacteria were grown for 36 h on BGA from standardized stocks, resuspended in PBS (pH 7.2) containing 500 μM of hypoxanthine, and then diluted in the same solution to obtain ~ 50,000 bacteria in 300 μl in 15-ml tubes. 0.05 U.ml$^{-1}$ xanthine oxidase was added to produce $H_2O_2$ and $O_2 \cdot ^{-}$ [29], and the bacteria were incubated for 30 min at 37 °C with shaking. Serial dilutions were plated onto BGA to count CFUs, using control suspensions incubated as above. Three biological replicates and three technical replicates were made. CFU counts of the initial suspensions were used as references.

**Construction of strains and plasmids.** Deletion mutants were obtained by allelic exchange[52]. Complementation was performed at the *ure* chromosomal locus[52]. The inactivation mutants were constructed using the suicide vector pFUS2[53]. For pProm1727-mRFP1, the *cueR-copZ* intergenic region including the first 7 codons of *copZ*, and the *mRFP1* gene without its initiation codon were amplified by PCR and cloned as BamHI-XbaI and XbaI-HindIII restriction fragments, respectively, to generate a translational fusion in pBBR1-MCS5[54]. All primers are listed in Supplementary Data S7.

**RNA sequencing and qRT-PCR.** For RNA extractions, the cultures were stopped by adding 2 mL of a mixture of phenol:ethanol (5:95) to 8 mL of bacterial suspensions. Bacteria were pelleted, and total RNA was extracted with TriReagent (InVitrogen)[55]. Genomic DNA was removed by a DNAse I treatment (Sigma Aldrich). DNA-depleted total RNA was treated with the RiboZero rRNA Removal kit (Illumina). The rRNA-depleted RNA was then used to generate the Illumina libraries using the TrueSeq RNA library, followed with sequencing on an Illumina NextSeq 500 benchtop sequencer on SR150 high output run mode. The RNA-seq data were analyzed using Rockhopper v2.0.3 with the default parameters to calculate the reads per kilobase per million base pairs (RPKM) values for each coding sequence using the *B. pertussis* Tohama I BX470248 genome annotation. All RNA-seq experiments were performed on biological duplicates. The results fully supported preliminary microarray analyses for *B. pertussis*. qRT-PCR were performed on 3 or 4 separate cultures with at least 3 technical replicates for each condition. To map the transcriptional initiation site by rapid amplification of cDNA 5' ends (RACE), total RNA obtained as described above was treated with the GeneRacer kit (Invitrogen) using appropriate RACE primers (Supplementary Data S7).

**Mass spectrometry proteomic analyses.** *B. bronchiseptica* and *B. pertussis* were grown in SS medium supplemented or not with 2 mM $CuSO_4$ for 10 h and 16 h, respectively, and the cultures were stopped at $OD_{600}$ of 1.6–1.8. The bacteria were collected by centrifugation, resuspended in 50 mM Tris-HCl (pH 7.2), 100 mM NaCl, with Complete protease inhibitor (Roche) and lysed with a French Press. Clarified lysates were ultracentrifuged for 1 h at 100,000 g at 4 °C to separate soluble and insoluble fractions. Both fractions were heated at 100 °C in 5% SDS, 5% β-mercaptoethanol, 1 mM EDTA, 10% glycerol, 10 mM Tris (pH 8) for 3 min and loaded on a 10% acrylamide SDS-PAGE gel. The migration was stopped soon after the samples entered the separating gel. The gel was briefly stained with Coomassie Blue, and one gel slice was taken for each sample. The gel plugs were washed twice in 25 mM $NH_4HCO_3$ (50 μL) and acetonitrile (50 μL), the Cys residues were reduced and alkylated by adding 50 μL dithiothreitol (10 mM solution) and incubating at 57 °C before the addition of 50 μL iodoacetamide (55 mM solution). The plugs were dehydrated with acetonitrile, and the proteins were digested overnight at room temperature using porcine trypsin (12.5 ng/50 μL) in 25 mM $NH_4HCO_3$. Tryptic peptides were extracted first with 60% acetonitrile and 5%

formic acid for 1 h, and then with 100% acetonitrile until dehydration of the gel plugs. The extracts were pooled and excess acetonitrile was evaporated before analysis. Peptide fractionation was performed using a 500-mm reversed-phase column at 55 °C and with a gradient separation of 170 min. Eluted peptides were analyzed using Q-Extractive instruments (Fisher Scientific)[56]. Raw data collected during nanoLC-MS/MS analyses were processed and converted into *.mgf peak list format with Proteome Discoverer 1.4 (Thermo Fisher Scientific). MS/MS data were interpreted using search engine Mascot (version 2.4.0, Matrix Science, London, UK) installed on a local server. Searches were performed with a tolerance on mass measurement of 10 ppm for precursor ions and 0.02 Da for fragment ions, against two target decoy databases composed of all potential open reading frames of at least 30 residues between stop codons of *B. pertussis* or *B. bronchiseptica* (54890 and 74418 entries, respectively), plus sequences of recombinant trypsin and classical contaminants (118 entries). Cys carbamidomethylation or propionamidation, Met oxidation, and protein N-terminal acetylation were searched as variable modifications. Up to one trypsin missed cleavage was allowed. Peptides were filtered out with Proline 2.0 according to the cutoff set for proteins hits with 1 or more peptides larger than 9 residues, ion score > 10 and 2% protein false positive rates[56]. Spectral counting analyses were performed with Proline 2.0. The p-values were calculated with a Student's T-test (confidence level: 95%). To obtain q-values, the p-values were adjusted for multiple testing correction to control the false discovery rate by following the Benjamini-Hochberg procedure (Q = 5%)[57].

To identify CopZ, *B. pertussis* extracts were separated by electrophoresis using commercial Tricine 16% gels (InVitrogen) and stained with colloidal Coomassie blue dye. The fast-migrating protein band was digested in the gel with trypsin, and the peptide fingerprints were obtained by matrix assisted laser desorption/ionization time-of-flight mass spectrometry.

**Protein production.** For the production of CopZ, PrxGrx and GorB, recombinant pQE30 derivatives (Qiagen) were introduced into *E. coli* M15(pREp4). The cells were grown in LB medium at 37 °C under orbital shaking, and expression was induced at $OD_{600}$ of 0.8 with 1 mM IPTG for 3 h. Production of recombinant OxyR was done in *E. coli* SG13009(pREp4) in M9 medium with 2% casaminoacids and 250 μM ascorbate. At $OD_{600}$ of 0.4, 100 μM IPTG was added, and the culture was continued for 16 h at 14 °C. The bacteria were collected, and the pellets were resuspended in 50 mM Tris-HCl (pH 7.5), 100 mM NaCl, 10 mM imidazole, plus 1 mM TCEP for CopZ and OxyR. The bacteria were lysed with a French press, and the lysates were clarified by centrifugation. The proteins were purified by chromatography on Ni$^{++}$ columns. Concentrations were measured using the BCA assay kit (ThermoFisher Scientific) for PrxGrx and GorB, and with the Qbit protein assay kit for CopZ and OxyR (ThermoFisher Scientific).

**Metal binding to CopZ.** Purified CopZ (30 μM) was incubated in 50 mM Tris-HCl (pH 7.5), 100 mM NaCl, 1 mM TCEP, and 100 mM EDTA for 30 min with stirring at room temperature to remove bound metal ions and then dialyzed against the same buffer without EDTA, using 3.5-kD cutoff membranes. The apo form of the protein precipitated in the absence of TCEP. Iron sulfate or copper chloride solutions were prepared in the same buffer containing 1 mM TCEP and 25 mM ascorbate. CopZ was mixed with 125 μM of either metal for 5 min at room temperature, the samples were then dialyzed against the same buffer and finally acidified with 6% nitric acid. Elemental Cu and Fe analyses were performed by ICP-AES (Agilent, model 5110) in axial mode. The wavelengths selected for Cu and Fe analyses were 327.395 nm and 238.204 nm, respectively. $^{63}Cu$ and $^{65}Cu$ concentrations were both analyzed to evidence potential interferences. To remove polyatomic interferences, a 100 mL min$^{-1}$ He flow was added at the collision reaction interface. Impurity of $^{129}Xe$ in the Ar gas used for generating the plasma was chosen as internal standard to detect any instrumental drift. External calibrations were performed from standard solution at 1 g L$^{-1}$ (Astasol), adequately diluted and acidified to be in the range of the unknown concentrations.

**Enzymatic assays.** Buffer exchange was performed to obtain the purified recombinant proteins in 0.1 M sodium/potassium phosphate (pH 7.6). Enzymatic activity of GorB was followed at 25 °C in the same buffer containing 0.3 mM NADPH, 8 mM GSH, and $H_2O_2$ at concentrations ranging from 100 μM to 2 mM. The latter was used to generate the corresponding concentrations of oxidized glutathione, before starting the reaction by adding the enzyme at a concentration of 0.5 nM[27]. The decrease of NADPH concentration was followed by measuring the absorbance of the solution at 340 nm. Activity of PrxGrx was detected in a coupled assay performed at 25 °C in the same buffer with 0.3 mM NADPH, 6 mM GSH, 0.25 nM GorB and 1 μM PrxGrx[28]. 4 mM $H_2O_2$ was added last as the substrate of the first reaction to generate the substrate of the second reaction, oxidized glutathione. However, as $H_2O_2$ also directly generates oxidized glutathione, the activity of PrxGrx was detected by the increased rate at which NADPH was consumed when both enzymes were present in the reaction relative to the reaction rate with GorB only. No enzymatic constants were determined because of the coupled reaction set up.

**EMSA.** Amplicons of 127 bp and 139 bp, corresponding to the *cueR-copZ* (IGR 1) and *copZ-prxgrx* (IGR 2) intergenic regions, respectively, were mixed at 1 μM with

0, 2 or 20 μM recombinant OxyR, in 50 mM Tris-HCl (pH 7.5), 100 mM NaCl, 10 mM MgCl$_2$, 1 mM TCEP. After 30 min at 37 °C, the samples were loaded onto a 6% acrylamide gel containing 25 mM Tris-HCl (pH 8.8), 200 mM glycine, 10 mM MgCl$_2$. Electrophoresis was performed on ice for 90 min at 25 mA in the same buffer. DNA was detected by ethidium bromide staining.

**Phagocytosis assays**. THP1 cells were cultured in RPMI medium with 10% fetal bovine serum (RPMI/FBS). They were then treated with 50 ng/mL phorbol 12-myristate 13-acetate (PMA, Sigma) for 24 h to induce differentiation into macrophages and washed in PBS before adding fresh medium. The bacteria were grown at 37 °C on BGA for 48 h, resuspended in PBS and incubated with the macrophages at a multiplicity of infection (MOI) of 100 for 30 min at 37 °C with 5% CO$_2$. The macrophages were then washed with PBS containing 100 μg/ml polymyxin B to eliminate extracellular bacteria. For the following incubation the concentration of polymyxin was lowered to 5 μg/ml in RPMI/FBS, and one and three hours later macrophages were washed with PBS and lysed with 0.1% saponine. Serial dilutions were plated onto BGA for CFU counting. The experiments were performed at least in biological triplicates.

**Fluorescence microscopy**. To detect intracellular bacteria, the THP1 cells were cultured as described above. RPMI/FBS was supplemented with 50 μM bathocuproine disulfonate (BCS) or 20 μM CuCl$_2$ during THP1 differentiation and for the following 12 h, and after washing the cells in RPMI, fresh RPMI/FBS was added 1 h before contact. Bacteria harboring the reporter plasmid pProm1727-mRFP1 were grown for 12–13 h in SS medium containing 50 μM BCS. They were collected by centrifugation, resuspended in PBS to an OD$_{600}$ of 1 and incubated with macrophages at a MOI of 100. After one hour of contact, extracellular bacteria were eliminated, RPMI/FBS supplemented with copper or BCS was added, and the incubation was continued for one hour at 37 °C. Finally, the cells were fixed with Formalin (Sigma) for 15 min at room temperature and washed three times with PBS. The cell nuclei and the plasma membranes were labeled with DAPI at 30 μg/ml (Sigma) and Wheat Germ Agglutinin Alexa Fluor ™ 488 Conjugate (Invitrogen) at 5 μg/ml for 10 min at room temperature. Cells were then washed twice with PBS. Images were captured by confocal microscopy with a spinning disk microscope Nikon/Gataca system csu-w1. The images were analyzed with the Fiji software[58]. All parameters were set to the same values on all images. The bacteria were identified and counted with the 3D Object Counter module, and the fluorescence intensity was calculated by evaluating the average gray levels of the red objects. Three images containing each between 200 and 266 detectable bacteria were analyzed for each condition.

For ATP7A labeling, THP1 cells were cultured as above. One million cells were placed on top of coverslips in each well of a 24-well plate. The cells were then treated for 24 h with PMA in RPMI/FBS as above, with PMA + LPS (500 ng/mL), or with PMA for 24 h and then placed in contact with bacteria at a MOI of 100 for 2 h. The cells were then centrifuged for 3 min at 300 g, and the supernatant was removed. The centrifugation step was necessary to pellet the non-differentiated, non-adherent control cells. Fixation was performed for 45 min at room temperature with the eBioscience Fixation buffer (ThermoFisher Scientic). The cells were then washed three times by centrifugation and addition of eBioscience Permeabilization buffer (ThermoFisher Scientic). They were then incubated for 35 min with 5% goat serum in the permeabilization buffer, followed by an incubation of 2 h with a monoclonal antibody against ATP7A (Invitrogen MA5-27720) diluted 600 folds. The cells were washed three times as above, before adding Alexa Fluor 488-AffiniPure Donkey Anti-Mouse IgG (Jackson: 715-545-151) diluted 1/500 in permeabilization buffer with 5% goat serum also containing DAPI as described above. After 1 h incubation, the cells were washed three times with PBS. Images were obtained using an Evos M5000 microscope with an apochromatic X60 Olympus objective. The final image is a projection of 100 Z stacks with 0.2-μm steps. All images were captured with the same exposure parameters. The fluorescence intensity of the cells was calculated with the Fiji software (ImageJ) in two dimensions.

**Animal experiments**. The bacteria were grown on BGA for 48 h and resuspended in sterile PBS to 10$^6$ bacteria per 20 μL. Female, 6-weeks-old JAX ™ BALB/ cByJ mice (Charles River) were anesthetized intraperitoneally with a mixture of ketamine, atropine, and valium and infected by intranasal inoculation with 10$^6$ bacteria. Groups of 3-5 animals per bacterial strain were sacrificed 3 h, and 4, 7, 14, or 21 days post-inoculation. Their lungs and naso-pharynxes were removed in a sterile manner and homogenized with an Ultra Thurax apparatus[59]. The suspensions were serially diluted in PBS and plated onto BGA for CFU counting. All the experiments were carried out in accordance with the guidelines of the French Ministry of Research regarding animal experiments, and the protocols were approved by the Ethical Committees of the Region Nord Pas de Calais and the Ministry of Research (agreement number APAFIS#9107 ± 201603311654342 V3).

**Bioinformatic analyses**. The regions upstream of CueR-regulated genes in *E. coli* and *Rubrivivax gelatinosus* (*copA* and *cueO*) were aligned, and Glam2 from the

MEME 5.1.0 suite[60] was used to define a pattern. This motif was sought on the genome of *B. pertussis* using Glam2scan. To identify homologs of PrxGrx, the NCBI nr database was searched with the HMMER software[61] for proteins carrying both Redoxin (Pfam: PF08534) and Glutaredoxin (Pfam: PF00462) domain signatures. The corresponding genes and the five upstream and downstream genes were retrieved from databases of prokaryotic DNA sequences found in the NCBI site (GenBank, RefSeq, and WGS).

**Statistics and Reproducibility**. Three to six independent biological samples were used to determine growth yields and survival to oxidative stress or macrophages. For the animal colonization experiments, 4 and 5 mice per time point were used for the wt and mutant bacteria, respectively. To determine the Cu/CopZ ratios, three individual samples were processed. For experiments using samples of small sizes, statistical tests were performed with the GraphPad Prism software using non-parametric two-tailed Mann-Whitney tests (confidence level 95%). For qRT-PCR 3 independent biological samples were used. Given that the *cueR* and *oxyR* KO bacteria grew poorly in the presence of copper or hydrogen peroxide, respectively, the error bars were large. No statistical analyses were performed in those cases. For quantification of the fluorescence of intracellular bacteria, large numbers of bacteria were used (>600). In that case, the sample sizes allowed to use a parametric two-tailed Student T-test (confidence level 95%; 1381 degrees of freedom) of the GraphPad Prism software.

**Reporting summary**. Further information on research design is available in the Nature Research Reporting Summary linked to this article.

## Data availability
The transcriptomic data have been deposited in the GEO repository of NCBI (GEO accession: GSE145049). The mass spectrometry proteomics data have been deposited in the ProteomeXchange Consortium via the PRIDE[62] partner repository with the dataset identifier PXD020900 and 10.6019/PXD020900. The data used to generate the graphs and charts can be found in Supplementary Data S8.

## Code availability
The code generated in this work to retrieve the genetic environment of genes of interest is available to readers without restriction. It can be accessed using the following link: https://github.com/rudantoine/GeneEnv.git.

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

## Acknowledgements

We thank Violaine Dubois for sharing unpublished data, Hervé Drobecq for mass fingerprinting analyses, and Blanche Daunou for preliminary work with recombinant proteins. A.R.-M. and G.R. acknowledge the support of doctoral fellowships from the University of Lille- Region Hauts-de-France and from the University of Lille, respectively. A. R.-M. also thanks the Fondation de la Recherche Médicale (FRM) for their support. This work was funded by the Institut National de la Santé et de la Recherche Médicale (INSERM) and the University of Lille. ICP-AES measurements were performed on the Chevreul Institute Platform (U-Lille/CNRS). The Region Hauts de France and the French government are acknowledged for co-funding this equipment.

## Author contributions

A.R.-M., R.A., and F.J.-D. designed the study. A.R.-M., J.C., S.S. G.R., J.-M.S., G.B., V.A., D.H., S.S.-D. performed the experiments. A.R.-M., R.A., and F.J.-D. analyzed the data. A.R.-M. and F.J.-D. wrote the manuscript. C.L. provided advise. All authors read and made corrections to the manuscript.

## Competing interests

The authors declare no competing interests.
