## [Peer Review File · Communications Biology]

Reviewers' Comments:

Reviewer #1:

Remarks to the Author:

The manuscript by Rivera-Millot et al reports on the copper-defense mechanisms of the human pathogen *B. pertussis*. The main conclusion of the authors is that copper resistance mainly depends on three genes, organized in one operon that encode for a CopZ-like Cu chaperone, a chimeric peroxiredoxin-glutaredoxin protein (PrxGrx) and a glutathione reductase (GorB). Other Cu resistance genes that are typically present in bacteria are either absent or are inactivated by IS sequences.

In general, the manuscript is well written and the authors' conclusions are supported by the experimental data. Nevertheless, there are some issues the authors should take care of for substantiating their claims and for clarifying some statements. These issues are listed below:

1. There are some conceptual issues that the authors should address by extending the discussion/description of their results:

- a.) *B. pertussis* appears to be highly Cu resistant (5mM, e.g. Fig. S1), yet the authors suggest that sequestration of Cu by CopZ is the main defense mechanism. Assuming that there is indeed no active export system, wouldn't this lead to a rapid saturation of CopZ? In particular, if the cellular concentrations of CopZ in *B. pertussis* are similar to the concentrations in other organisms (1-10 μ M). This should be discussed.
- b.) The authors state in lane 113 that *B. pertussis* is unable to synthesize glutathione, but glutathione is involved in Cu tolerance and electron donor/acceptor for PrxGrx and GorB. I assume that *B. pertussis* engages glutathione uptake systems, but this is not mentioned.

2. There are also some experimental issues that should be addressed:

- a.) The authors determine the Cu stoichiometry of recombinant CopZ by ICP-MS and found a 3:1 ratio. As CopZ tends to dimerize, can the authors exclude that the 3:1 ratio actually reflect the monomer-dimer equilibrium, rather than Cu binding to the His-tag? This could be easily determined by ICP-MS and native PAGE after His-tag cleavage.
- b.) The authors contribute the extended lag phase of *B. pertussis* to the re-oxidation of Cu(I) by oxygen. If this is true, then the addition of ascorbate at the end of the lag phase should prevent growth. Alternatively, adding no ascorbate to the medium should result in no growth at all. These are easy to test conditions.
- c.) The silver-stained gel in Fig. S3 would be much more conclusive if also a strain over-expressing CopZ would be shown. Right now it is difficult to determine whether the indicated band indeed reflects CopZ or any other small Cu-induced protein.
- d.) The data on intracellular survival of the triple knock-out are intriguing, but here the authors could make a much stronger case if they would also look at the CopZ deletion versus the PrxGrx/GorB deletion. This would allow to differentiate between resistance to Cu and resistance to peroxide within macrophages.
- e.) Fig. 3E, left panel, requires a bright field image, otherwise it is difficult to determine whether the starting numbers of bacteria are the same.

3. Minor/editorial issues:

- a.) There is an imbalance between the figures in the main manuscript and the supplementary data. I would include Figures S1, S3, S7e into the main body of the manuscript.
- b.) If there is any explanation for the difference in lung and nasopharynx colonization. If so, this should be mentioned.
- c.) Lane 356: CO2
- d.) Lane 204: the term 'professional' is ambiguous.

- e.) The different grey scales in the bar charts throughout the manuscript are difficult to differentiate.
- f.) The label of the scale bar in Fig. 3e is difficult to read.
- g.) The qRT-PCR data only show that *prxgrx* is regulated in response to Cu by CueR and possibly by another regulator. The statement that the entire operon is Cu-regulated is likely true but not validated by the data shown in Fig. 3a-c.
- h.) Is there any explanation that the *copA* transcript increases upon Cu supplementation, but the protein decreases (Fig. 1b)? Also the spectral counts for PcoA and PcoB are lower upon Cu supplementation.

Reviewer #2:

Remarks to the Author:

The manuscript by Rivera-Millot et al. describes *B. pertussis* defense mechanism against cooper. They show that, unlike other bacteria, *B. pertussis* cooper defense consist only of a small number of proteins. They describe three proteins encoded by the bp1727-bp1728-bp1729 operon that have complementary activities against cooper and peroxide exposure. They found the expression of these proteins regulated by cooper as well as the presence of peroxides (bp1728 and bp1729) through two different transcriptional activators. The authors hypothesize that this streamlined defense against copper intoxication is due to the fact that *B. pertussis* does not have to cope with cooper intoxication in its natural environment, the human being, except when it is phagocytosed by the host cells

The manuscript is very interesting and provides new insights into the poorly understood biology of *B. pertussis*. There are a few points, however, that might need revision.

1. According to Mat & Met, to monitor copper influence on gene transcription by qRT-PCR and protein expression by proteomics, both *B. pertussis* and *B. bronchiseptica* were grown for 16 h in the presence of 2 mM CuSO₄ and the cultures were stopped at OD₆₀₀ of 1.5-1.6. However, according to Fig.S1 after 16 hours of cultivation only *B. bronchiseptica* (and in the absence of CuSO₄) might reach the OD₆₀₀ of 1.5-1.6. *B. pertussis* would still be in lag phase after 16 hs (FigS1 a). The authors should clarify this point.
2. Also regarding the conditions in which the samples were obtained, the Figure legends should state clearly how the samples were obtained to avoid confusion. For example, the different panels of Fig.1 show data obtained with bacteria treated in different ways (incubated with cooper during either 30 minutes or 16 hours). The reader has to search in the Results section to know how the results depicted in panels a) and b) were obtained. It would be very useful to have this kind information available in every figure legend.
3. In several parts of the manuscript the authors describe *B. pertussis* grown in the presence of CuSO₄ for 16 hours as "B. pertussis grown continuously in the presence of CuSO₄". The wording is misleading because a continous culture is a culture system completely different from the batch system used in this study. I would suggest to avoid the word "continuously" for clarity reasons.
4. In Fig S1 *B. pertussis* grown in SS reached a maximum OD₆₀₀ of around 1.5-1.6 at the stationary phase. However, panels a) and b) of Fig 2 show an OD₆₀₀ of around 4 for *B. pertussis* grown in SS during 24 hours. According to the Fig. 2 legend, panels a) and b) depict growth yield. What the authors mean by growth yield and how they calculate it should be clear in the manuscript (both in Mat& Met and in this Figure Legend)
5. Supplementary Figure S3 should be replaced by a westernblot.
6. The Tables depicting proteomic data (S4 and S5) should show the fold change of each protein under the different conditions along with the respective statistical significance

7. The data shown in Figure S7 should be part of the manuscript instead of supplementary material. The results shown in this Figure together with the model proposed (based on those results) are a very important part of this manuscript. The different studies whose results are shown in this Figure as well as the methodology used should be thoroughly described to give further support to the findings.

8. Figure 3 (panel e) shows the *copZ-prxgrx-gorB* operon response to the level of intracellular cooper. The Figure shows the copper-mediated upregulation of this operon in the intracellular bacteria. The authors mention that these bacteria are in phagolysosomes inside the cell. I agree that, by this confocal study they can conclude that the bacteria are intracellular. However, to be able to say that those bacteria are in phagolysosomes they should demonstrate it by colocalization of the bacteria with specific markers for lysosomes. In fact, previous studies (cited in this manuscript, Ref 38 and 39) have shown that upon phagocytosis by THP1-cells a significant number of bacteria inhibits phagolysosomal fusion, remaining alive for days within compartments with early endosomal characteristics. These previous reports further showed that only the bacteria outside phagolysosomes are alive inside the cell. So, the location of the bacteria shown in Fig 2 might be relevant to further dissect whether this operon is indeed needed for intracellular survival since it was previously shown that *B. pertussis* trafficked to lysosomes is actually killed. So, the real question is whether *B. pertussis* needs this cooper-resistant system within the compartments in which it remains alive inside the cells. In a previous study on the proteomic evolution of intracellular *B. pertussis* inside THP1 the proteins described in this study were not found (Journal of Proteomics 136 (2016) 55–67). However, not finding a protein in a proteomic study is not conclusive since they might be difficult to detect because of the size, the abundance, etc. Summarizing, the authors cannot claim that the bacteria in Figure 2 are in phagolysosomes unless they check the location of the intracellular bacteria using specific markers for lysosomes. On the other hand, if they confirm that the bacteria in Fig 2 are in lysosomes the authors need to discuss how this cooper defense system would help a bacteria residing in a compartment in which they will be killed by other bactericidal mechanisms apart from cooper and peroxide.

Reviewer 1. We thank the reviewer for their positive assessment of our work.

Comment

a.) *B. pertussis* appears to be highly Cu resistant (5mM, e.g. Fig. S1), yet the authors suggest that sequestration of Cu by CopZ is the main defense mechanism. Assuming that there is indeed no active export system, wouldn't this lead to a rapid saturation of CopZ? In particular, if the cellular concentrations of CopZ in *B. pertussis* are similar to the concentrations in other organisms (1-10 μ M). This should be discussed.

Response

The reviewer is right that the apparently high resistance of *B. pertussis* is not consistent with the absence of an active extrusion system. Actually, *B. pertussis* is not very resistant to copper, but its standard culture medium used in this study, the Stainer-Scholte medium, contains large amounts of glutamate, casamino acids, Tris, glutathione, and cysteine, all of which can chelate copper and thus reduce its effective concentration. We thus added this explanation for the apparent resistance of *B. pertussis* to copper (lines 80-83).

“The apparent resistance of both species to high concentrations of copper is misleading and due to growth medium compounds chelating the metal ion, in particular Tris, amino acids and glutathione, which markedly reduces its effective concentration”

In addition, the oxidizing culture conditions of *B. pertussis*, which is a strictly aerobic organism growing in strongly aerated cultures, result in the accumulation of less toxic Cu(II) (lines 83-87).

“We hypothesized that *B. pertussis* started growing after the toxic Cu¹⁺ ion resulting from the reduction of Cu²⁺ by ascorbate, another component of the growth medium, was re-oxidized thanks to strong oxygenation of the cultures. This was confirmed by the observation that the addition of fresh ascorbate after a few hours to *B. pertussis* cultured in the presence of 5 mM copper prevented bacterial growth (Supplementary Fig. S1).”

Comment

b.) The authors state in lane 113 that *B. pertussis* is unable to synthesize glutathione, but glutathione is involved in Cu tolerance and electron donor/acceptor for PrxGrx and GorB. I assume that *B. pertussis* engages glutathione uptake systems, but this is not mentioned.

Response

Indeed, *B. pertussis* actively imports glutathione. This was added to the text of the revised version (lines 122-125).

“*B. pertussis* cannot synthesize glutathione but actively imports it, as indicated by the high expression levels of the glutathione-specific ABC transporter genes *bp3828-3831* in all conditions (Supplementary Tables S1 and S2).”

Comment

2a.) The authors determine the Cu stoichiometry of recombinant CopZ by ICP-MS and found a 3:1 ratio. As CopZ tends to dimerize, can the authors exclude that the 3:1 ratio actually reflect the monomer-dimer equilibrium, rather than Cu binding to the His-tag? This could be easily determined by ICP-MS and native PAGE after His-tag cleavage.

Response

The goal of this experiment was to show that CopZ binds copper, which is demonstrated by the difference between Cu and Fe, the latter being used as a control. The experiments proposed by the reviewer are difficult to perform because the *B. pertussis* recombinant CopZ readily precipitates in vitro. The Fe control was used to determine the stoichiometry. Indeed, upon incubation of the apoprotein with Fe or Cu, one ion was chelated the 6-His tag used for CopZ purification. Therefore, we subtracted a Cu ion per monomer to the value obtained by ICP-AES, yielding a ratio of 2 Cu per CopZ. This value is consistent with crystal structures of homologues showing 4 Cu ions bound at the interface between two monomers, as is now explained in the text (lines 131-139). We also added Fig. S6 to show the sequence conservation of CopZ and the Cu-binding mode of a CopZ homologue, which comfort our results.

“Inductively Coupled Plasma Atomic Emission Spectroscopy (ICP-AES) analyses were performed with purified CopZ incubated with Cu or with Fe as a control. Three Cu ions and one Fe ion were bound per CopZ monomer, respectively (Fig. 5a). We reasoned that in both cases one ion was chelated by the 6-His tag used for recombinant CopZ purification. Therefore, subtraction of a Cu ion yields a ratio of 2 Cu per CopZ monomer. This value is consistent with crystal structures of homologues (pdb accession numbers 6FF2 and 2QIF) showing CopZ dimerization mediated by 4 Cu ions bound by conserved Cys and His residues²⁵ (Supplementary Fig. S6). Thus, CopZ is a copper binding protein, which in the absence of a functional CopA in *B. pertussis* most likely sequesters cytosolic copper to counter its toxicity.”

Comment

2b.) The authors contribute the extended lag phase of *B. pertussis* to the re-oxidation of Cu(I) by oxygen. If this is true, then the addition of ascorbate at the end of the lag phase should prevent growth. Alternatively, adding no ascorbate to the medium should result in no growth at all. These are easy to test conditions.

Response

The experiment with ascorbate proposed by the reviewer was performed and confirmed our explanation for the bacteriostatic effect of copper (Fig. S1), as explained lines 83-87:

“We hypothesized that *B. pertussis* starts growing after the toxic Cu¹⁺ ion resulting from the reduction of Cu²⁺ by ascorbate, another component of the growth medium, was re-oxidized thanks to strong oxygenation of the growth medium. This was confirmed by the observation that the addition of fresh ascorbate after a few hours to *B. pertussis* cultured in the presence of 5 mM copper prevented bacterial growth (Supplementary Fig. S1).”

Comment

2c) The silver-stained gel in Fig. S3 would be much more conclusive if also a strain over-expressing CopZ would be shown. Right now it is difficult to determine whether the indicated band indeed reflects CopZ or any other small Cu-induced protein.

Response

Figure 3. Production of CopZ by *B. pertussis*. (a) Lysates of wild type *B. pertussis* (wt) or the deletion mutant (Δ bp1727) grown in SS medium supplemented (+ Cu) or not with 2 mM CuSO₄ were subjected to SDS-PAGE electrophoresis in Tris-tricine gels for the detection of CopZ. The gel was stained with colloidal Coomassie blue dye. CopZ migrates below the 10 kDa band of the markers (ladder) in the wt + Cu lane, whereas it is absent from the deletion mutant. (b and c) The protein band was cut from the gel, and CopZ was identified by mass fingerprinting analyses. The m/z ratios of the peptides identified by MALDI/TOF are shown in panel b, and the sequences of the peptides are in panel c. This experiment complements the proteomic analyses, because the small size of CopZ hampered its detection in global proteomic approaches (see Fig. 2b and Supplementary Table S4).

We performed this experiment using colloidal Coomassie staining and mass fingerprinting analyses to confirm that CopZ is overproduced in response to Cu. This is now shown in Fig. 3 and described in the text (lines 101-104)

“Because the small size of CopZ might have hampered its detection in global proteomic analyses, we analyzed extracts of copper-treated *B. pertussis* by denaturing electrophoresis and mass fingerprinting of the CopZ band (Fig. 3). This confirmed that CopZ is strongly overproduced by *B. pertussis* in the presence of copper.”

Comment

2d) The data on intracellular survival of the triple knock-out are intriguing, but here the authors could make a much stronger case if they would also look at the CopZ deletion versus the PrxGrx/GorB deletion. This would allow to differentiate between resistance to Cu and resistance to peroxide within macrophages.

Response

We determined the survival of *B. pertussis* in macrophages using the mutants suggested by the reviewer, namely the *copZ* KO mutant and the *prxgrx-gorB* KO mutant. The results are presented in Fig. S7. They show that both CopZ and Prxgrx-GorB contribute to survival in macrophages, which is fully consistent with their functions. This is reported lines 158-161.

Supplementary Figure S7. Intracellular survival of *B. pertussis* in THP1 macrophages. The survival rates of the *copZ* KO mutant ($\Delta 1727$) and the *prxgrx-gorB* KO mutant ($\Delta 1728-29$) were compared with those of the parental strain and of the full deletion mutant ($\Delta 1727-28-29$). Three replicates were made for each strain. The means and standard deviations are shown.

“We also subjected the *copZ* and *prxgrx-gorB* mutants to this assay. Both were killed faster than the wt strain in macrophages, especially at the 1-hour time point. This shows that both CopZ and the Prxgrx-GorB system contribute to survival early after phagocytosis (Supplementary Fig. S7).

Comment

2e) Fig. 3E, left panel, requires a bright field image, otherwise it is difficult to determine whether the starting numbers of bacteria are the same.

Response

We added new panels to the left of the figure showing the projection of the bacteria as objects in the fields. In this manner, one can appreciate that similar numbers of bacteria are present in both cases.

Comment

3a) There is an imbalance between the figures in the main manuscript and the supplementary data. I would include Figures S1, S3, S7e into the main body of the manuscript.

Response

We agree with the reviewer that some supplementary figures should be in the main manuscript, and we modified the manuscript in this sense. There are now 7 figures in the main text.

Former Fig. S1 is now Fig. 1

Fig. S3 is Fig. 3

As also suggested by Reviewer 2, Fig. S7 became Fig. 4. Its legend was shortened, and the experimental details can be found in the Methods section.

We also split former Fig. 3 in two separate figures for the sake of clarity. Fig. 6 now pertains to in vitro regulation of the operon, and Fig. 7 shows the activation of the operon in macrophages.

Comment

3b) If there is any explanation for the difference in lung and nasopharynx colonization. If so, this should be mentioned.

Response

We have no strong explanation for the difference between the two organs. However, work in the lab has shown that there is an influx of immune cells in the nose over the course of the infection, which might contribute to explain the difference between WT and mutant strains in that organ, as briefly discussed lines 261-265.

“Absence of the *copZ-prxgrx-gorB* operon had little effect on *B. pertussis* colonization of the lungs but significantly shortened the colonization of the nasal cavity of mice. Interestingly, in murine nasal tissues a large influx of immune cells occurs after one to two weeks of infection, and a significant proportion of the bacteria appear to reside within immune cells in that organ (V. Dubois, personal communication). This might contribute to explain the negative effect of the mutation on nasal cavity colonization.”

Comment

3c) Lane 356: CO2

Response

The misspelling was corrected. Thanks for spotting it.

Comment

3d) Lane 204: the term ‘professional’ is ambiguous

Response

We agree with the reviewer and replaced ‘professional’ with ‘genuine’.

Comment

3e) The different grey scales in the bar charts throughout the manuscript are difficult to differentiate.

Response

The shades of grey represent the same strains in all panels (Fig. 4), and the order of the bars is the same as the order in the visual legend, which should make the figure readable.

Comment

3f) The label of the scale bar in Fig. 3e is difficult to read.

Response

It was enlarged.

Comment

3g) The qRT-PCR data only show that *prxgrx* is regulated in response to Cu by CueR and possibly by another regulator. The statement that the entire operon is Cu-regulated is likely true but not validated by the data shown in Fig. 3a-c.

Response

As explained in the text, qRT PCR was unsuccessful on the *copZ* gene, because no suitable pair of oligonucleotides was found (lines 169-170). However, transcriptomic data (Fig. 2 and Tables S1 and S2) show that the three genes are regulated by copper. In addition, RNASeq analyses (Fig. S2) show that the three genes form an operon. As 1° Cu-induced upregulation of the second gene of the operon is clearly affected in the *cueR* KO mutant, 2° the first gene is also regulated by Cu, and 3° the three genes form an operon, it is very reasonable to assume that the first gene is also regulated by CueR. Our reasoning was explained with greater detail in the text (lines 173-178).

“The three genes are co-transcribed (Supplementary Fig. S2) and up-regulated by copper (Fig. 2 and Supplementary Tables S1-S2), and we showed that the second is controlled by CueR. We can thus reasonably assume that the entire operon is under the control of CueR. Interestingly, though, the remaining copper-induced up-regulation of *prxgrx* in the *cueR* KO mutant indicated that the operon may also be controlled by a second system directly or indirectly activated by copper.”

Comment

3h) Is there any explanation that the *copA* transcript increases upon Cu supplementation, but the protein decreases (Fig. 1b)? Also the spectral counts for PcoA and PcoB are lower upon Cu supplementation.

Response

As pointed out by the reviewer 2, statistical values were required for the proteomic experiments. We thus performed those experiments in triplicate with independent biological samples for the two species in two different conditions (with or without Cu). For practical reasons, the new proteomic experiments were performed on a single band rather than 5 for each sample, and therefore fewer spectra were obtained for each protein. With the new sets of data, the effects mentioned by the reviewer are no longer apparent. We believe that they were mostly due to the fact that *pcoA* and *pcoB* are poorly transcribed in *B. pertussis*, and that *copA* is interrupted by an IS, as explained in the text. The discrepancy between transcriptomic and proteomic data might have been caused by the low levels of expression of the three genes in *B. pertussis*.

Reviewer 2. We also thank this reviewer for their very positive assessment of our work.

Comment

- 1) According to Mat & Met, to monitor copper influence on gene transcription by qRT-PCR and protein expression by proteomics, both *B. pertussis* and *B. bronchiseptica* were grown for 16 h in the presence of 2 mM CuSO₄ and the cultures were stopped at OD₆₀₀ of 1.5-1.6. However, according to Fig.S1 after 16 hours of cultivation only *B. bronchiseptica* (and in the absence of CuSO₄) might reach the OD₆₀₀ of 1.5-1.6. *B. pertussis* would still be in lag phase after 16 hs (FigS1 a). The authors should clarify this point.

Response

The growth curves shown in Fig S1 (now Fig 1) were performed with 5 mM copper in the culture medium, whereas the transcriptomic analyses were performed with cultures containing 2 mM copper. At the latter concentration, there is no significant lag in the cultures.

In addition, the turbidities of the cultures as measured for the growth curves cannot be directly compared with the OD₆₀₀ used in the other experiments as two different apparatus were used. Additional details on the growth conditions were added in the Mat and Met section (lines 299-308):

“To record growth curves, the optical density at 630 nm was continuously measured using an Elocheck device (Biotronix). For growth yields, the *B. pertussis* cultures were started at OD₆₀₀ of 0.1, and after 24 h of growth OD₆₀₀ were determined using an Ultrospec 10 spectrophotometer (Biochrom) with five biological replicates for each condition and strain. Glutathione was omitted from the culture medium where indicated. The turbidities measured using the Elocheck and Ultrospec apparatus cannot be directly compared, as they use different wavelengths and pathlengths. For transcriptomic, proteomic and qRT-PCR experiments, the *B. pertussis* cultures were started at OD₆₀₀ of 0.1 to reach OD₆₀₀ values of 1.5 after 16-20 h, and *B. bronchiseptica* cultures were started at OD₆₀₀ of 0.08 to reach OD₆₀₀ values of 1.5-1.8 after 10-12 h.”

Comment

- 2) Also regarding the conditions in which the samples were obtained, the Figure legends should state clearly how the samples were obtained to avoid confusion. For example, the different panels of Fig.1 show data obtained with bacteria treated in different ways (incubated with copper during either 30 minutes or 16 hours). The reader has to search in the Results section to know how the results depicted in panels a) and b) were obtained. It would be very useful to have this kind information available in every figure legend

Response

We provided more information in the legends:

“Figure 2. Copper regulation of homeostasis systems in *B. pertussis* and *B. bronchiseptica*. (a, c and d) RNAseq analyses of *B. pertussis* grown for 16 hours in the presence of 2 mM CuSO₄ in SS medium (a), *B. pertussis* grown in SS medium and treated for 30 minutes with 2 mM of CuSO₄ (c), and *B. bronchiseptica* grown for 10 to 12 hours in the presence of 2 mM CuSO₄ (d). Comparisons were made with bacteria grown in standard conditions. Each gene is represented by a dot. The x and y axes show absolute levels of gene expression in reads per kilobase per million base pairs (RPKM) in standard and copper conditions, respectively. The genes indicated in blue have the strongest regulation. The full sets of data are shown in Supplementary Tables S1, S2 and S3. (b) Summary of the transcriptomic and proteomic analyses performed after growing bacteria as above in medium supplemented with 2 mM CuSO₄. Standard culture conditions were used for comparisons.”

Comment

3) In several parts of the manuscript the authors describe *B. pertussis* grown in the presence of CuSO₄ for 16 hours as “*B. pertussis* grown continuously in the presence of CuSO₄”. The wording is misleading because a continuous culture is a culture system completely different from the batch system used in this study. I would suggest to avoid the word “continuously” for clarity reasons.

Response

The reviewer is correct that our wording was ambiguous. Instead of continuously, we stated the duration of the exposure to copper.

Comment

4) In Fig S1 *B. pertussis* grown in SS reached a maximum OD₆₀₀ of around 1.5-1.6 at the stationary phase. However, panels a) and b) of Fig 2 show an OD₆₀₀ of around 4 for *B. pertussis* grown in SS during 24 hours. According to the Fig. 2 legend, panels a) and b) depict growth yield. What the authors mean by growth yield and how they calculate it should be clear in the manuscript (both in Mat& Met and in this Figure Legend)

Response

The reviewer is right that the very different turbidities measured with the Elocheck apparatus and with a standard spectrophotometer are confusing. The values (absorbency units) obtained with an Elocheck do not correspond to optical density measurements classically obtained with a spectrometer, since the Elocheck instrument uses distinct pathlength and wavelength. This was explained in the figure legends and methods section (lines 298-304). We also modified the ordinate for the measurement values given by the Elocheck apparatus.

“To record growth curves, the optical density at 630 nm was continuously measured using an Elocheck device (Biotronix). For growth yields, the *B. pertussis* cultures were started at OD₆₀₀ of 0.1, and after 24 h of growth OD₆₀₀ were determined using an Ultrospec 10 spectrophotometer (Biochrom) with five biological replicates for each condition and strain. Glutathione was omitted from the culture medium where indicated. The turbidities measured using the Elocheck and Ultrospec apparatus cannot be directly compared, as they use different wavelengths and pathlengths.”

“Figure 4. Role of the operon in *B. pertussis* and in host-pathogen interactions. (a and b) Growth yields of *B. pertussis* after 24 h in SS medium (a) or in SS medium devoid of glutathione (b), supplemented or not with 2 mM CuSO₄. The cultures were inoculated at initial OD₆₀₀ values of 0.1, and after 24 h the OD₆₀₀ were determined....”

Comment

5) Supplementary Figure S3 should be replaced by a westernblot

Response

Given the time necessary to generate a serum, we felt that the best way to solve this question was to perform mass fingerprinting analyses of the protein band, as explained in the responses to reviewer 1 (point 2c).

Comment

- 6) The Tables depicting proteomic data (S4 and S5) should show the fold change of each protein under the different conditions along with the respective statistical significance

Response

In order to provide statistical significance to the proteomic data, the analyses were performed on three independent samples for each species and condition (*B. pertussis* in standard conditions; *B. pertussis* in the presence of copper; *B. bronchiseptica* in standard conditions; *B. bronchiseptica* in the presence of copper). However, for practical reasons, they were performed with one band (rather than 5) for each sample. The initial data were excluded from the analysis as they were obtained in a different manner, and statistical analyses were performed on the new data sets. Thus, the proteomic data now shown in Fig. 2 are based on the new analyses. Tables S4 and S5 now contain the requested statistical analyses.

Comment

- 7) The data shown in Figure S7 should be part of the manuscript instead of supplementary material. The results shown in this Figure together with the model proposed (based on those results) are a very important part of this manuscript. The different studies whose results are shown in this Figure as well as the methodology used should be thoroughly described to give further support to the findings.

Response

We followed the reviewer's advice and incorporated Fig. S7 in the main text. It is now Fig. 5. The necessary experimental details were provided in the Material and Methods section.

Comment

- 8) Figure 3 (panel e) shows the *copZ-prxgrx-gorB* operon response to the level of intracellular copper. The Figure shows the copper-mediated upregulation of this operon in the intracellular bacteria. The authors mention that these bacteria are in phagolysosomes inside the cell. I agree that, by this confocal study they can conclude that the bacteria are intracellular. However, to be able to say that those bacteria are in phagolysosomes they should demonstrate it by colocalization of the bacteria with specific markers for lysosomes. In fact, previous studies (cited in this manuscript, Ref 38 and 39) have shown that upon phagocytosis by THP1-cells a significant number of bacteria inhibits phagolysosomal fusion, remaining alive for days within compartments with early endosomal characteristics. These previous reports further showed that only the bacteria outside phagolysosomes are alive inside the cell. So, the location of the bacteria shown in Fig 2 might be relevant to further dissect whether this operon is indeed needed for intracellular survival since it was previously shown that *B. pertussis* trafficked to lysosomes is actually killed. So, the real question is whether *B. pertussis* needs this copper-resistant system within the compartments in which it remains alive inside the cells. In a previous study on the proteomic evolution of intracellular *B. pertussis* inside THP1 the proteins described in this study were not found (Journal of Proteomics 136 (2016) 55–67). However, not finding a protein in a proteomic study is not conclusive since they might be difficult to detect because of the size, the abundance, etc. Summarizing, the authors cannot claim that the bacteria in Figure 2 are in phagolysosomes unless they check the location of the intracellular bacteria using specific markers for lysosomes. On the other hand, if they confirm that the bacteria in Fig 2 are in lysosomes the authors need to discuss how this copper defense system would help a bacteria residing in a compartment in which they will be killed by other bactericidal mechanisms apart from copper and peroxide.

Response

The reviewer raises an important point here, and we thank them for bringing up this issue.

The use of the word ‘phagolysosomes’ was unfortunate from our part, especially as our data suggest that the CopZ-Prxgrx-GorB system is activated early and appears to be important at early time points after phagocytosis. Therefore, we removed the term ‘phagolysosomes’ from the text. We agree with the reviewer that the system under study most likely enhances survival in early endosomes. This is consistent with all our data.

As a complement of response, we labelled the Cu-specific transporter ATP7A in THP1 cells, which is shown in Fig. S9. We thus showed that differentiation into macrophages causes a strong upregulation of the expression of ATP7A in THP1 cells, with a punctuated labelling indicating its re-localization to cytosolic vesicles. ATP7A labeling was however too diffuse to allow co-localization with bacteria. Mobilization of the Cu transporter to endosomes (Fig. S9) is consistent with our observations that bacteria become fluorescent early after phagocytosis (Fig. 7), implying a rapid induction of the operon in macrophages replete with copper. Thus, our observations are in good agreement with the idea that the CopZ-PrxGrx-GorB system is induced early and might enhance survival of *B. pertussis* within early endosomes. This is also consistent with the observations that the absence of the operon affected survival at early time points after phagocytosis (Figs. 4d and S7). The new experiments and their interpretations are described lines 185-201.

“We then turned to human macrophages to determine if the operon is up-regulated upon phagocytosis. It was shown earlier that the ATP7A Cu-transporting ATPase is overexpressed and localizes to phagosomes in activated murine macrophages {White, 2009 #109}. We thus reasoned that copper might trigger expression of the *copZ-prxgrx-gorB* operon in *B. pertussis* early after phagocytosis. By using epifluorescence microscopy, we showed that differentiation and activation of THP1 cells cause overproduction of ATP7A. Upon differentiation, the Cu transporter partially redistributes from a mainly ER/Golgi localization in resting cells to a more punctuated localization likely corresponding to cytosolic vesicles (Supplementary Fig. S9). Early endosomes are thus likely to be loaded with copper. Then, to visualize the activation of the operon in intracellular bacteria, we introduced a transcriptional fusion between the promoter of the operon and an mRFP1 reporter gene in *B. pertussis* and put the bacteria in contact with differentiated THP1 cells deprived of copper or supplemented with copper. After one hour of contact, fluorescence of bacteria engulfed by copper-supplemented macrophages was significantly more intense than in copper-deprived macrophages (Fig. 7a and b). The rapid induction of the *copZ-prxgrx-gorB* promoter indicates that copper is an early signal for intracellular bacteria, and that CopZ-Prxgrx-GorB serves as a defense system in early phagosomes. This is supported by the role of that system for survival at early time points after phagocytosis (Fig. 4d).”

Supplementary Figure S9. Detection of the Cu transporter ATP7A in THP1 cells. (a) Representative fluorescence microscopy images of THP1 cells labelled for the ATP7A Cu transporter (in green), with the nuclei stained with DAPI (blue). The labelling was performed on non-differentiated monocytes (\emptyset), on monocytes differentiated for 24 hours into macrophages with PMA (PMA), with PMA and LPS (PMA + LPS), or with PMA before 2 hours of contact with *B. pertussis* (PMA + Bp). (b) Levels of green fluorescence (arbitrary units) of the cells labeled for ATP7A in the four conditions. Statistical analyses were performed using a non-parametric two-tailed Kruskal-Wallis test with Dunn's post-test (****, $p < 0.0001$).

Reviewers' Comments:

Reviewer #1:

Remarks to the Author:

The authors have sufficiently addressed my previous concerns and present now an interesting and important study.

Reviewer #2:

Remarks to the Author:

The authors have addressed all the issues raised. In my opinion the article improved considerably and may be accepted for publication